# Beyond the limitation of a single query: Train your LLM for query expansion with Reinforcement Learning

## Abstract

Reasoning-augmented search agents, such as Search-R1, are trained to reason, search, and generate the final answer iteratively. Nevertheless, due to their limited capabilities in reasoning and search, their performance on multi-hop QA benchmarks remains far from satisfactory. To handle complex or compound queries, we train an LLM-based search agent with the native capability of query expansion through reinforcement learning. In each turn, our search agent proposes several query variants, which are searched simultaneously to cover more relevant information. Meanwhile, given limited post-training data and computing resources, it is very challenging for a search agent to master multiple tasks, including query generation, retrieved information understanding, and answer generation. Therefore, we propose incorporating a pre-trained squeezer model that helps the search agent understand the retrieved documents, allowing the search agent to focus on query generation for high retrieval recall. With the assistance of the squeezer model, we discover that even a small-scale 3B LLM can demonstrate a strong capability of query expansion and achieve state-of-the-art accuracy on the multi-hop QA benchmarks. To be specific, our experiments across seven question-answering benchmarks demonstrate that our method, named *ExpandSearch*, achieves an average improvement of $4.4\%$ compared to state-of-the-art baselines, with strong gains on multi-hop reasoning tasks requiring diverse evidence aggregation.

## 1 Introduction

Large Language Models (LLMs) (Achiam et al., 2023; Team et al., 2023; Dubey et al., 2024; Yang et al., 2024) augmented with reasoning and search capabilities have shown remarkable progress in tackling complex information retrieval tasks that require multi-step reasoning over external knowledge sources. These reasoning-augmented search agents dynamically query databases and process retrieved information to overcome the inherent limitations of static parametric knowledge (Gao et al., 2023). Recent advances in reinforcement learning with verifiable rewards (RLVR) (Guo et al., 2025) have enabled these agents to learn search strategies, decompose complex questions, and iteratively refine their information gathering process (Jin et al., 2025b; Zhao et al., 2025).

However, existing search agents face two challenges, limiting their effectiveness in complex reasoning scenarios. First, they generate *semantically impoverished queries* that fail to capture the full spectrum of relevant information aspects (Carpineto & Romano, 2012). When confronted with multi-faceted questions requiring diverse evidence, these agents may produce narrow queries that miss crucial semantic information essential for comprehensive understanding. Meanwhile, in each turn, the search agent generates a single query and the single-vector embedding-based retrieval suffers from theoretical limitations as discussed in Weller et al. (2025). Second, they suffer from *information overload*, where the retrieved content contains substantial irrelevant information that obscures critical facts and degrades reasoning quality (Liu et al., 2024; An et al., 2024). This noise-to-signal ratio problem becomes particularly acute in multi-hop reasoning tasks where agents must navigate through extensive search results to identify sparse but crucial evidence. Meanwhile, as the context for the LLM, the lengthy retrieved content would take a large amount of computational costs and GPU memory, which significantly slows down the training process.

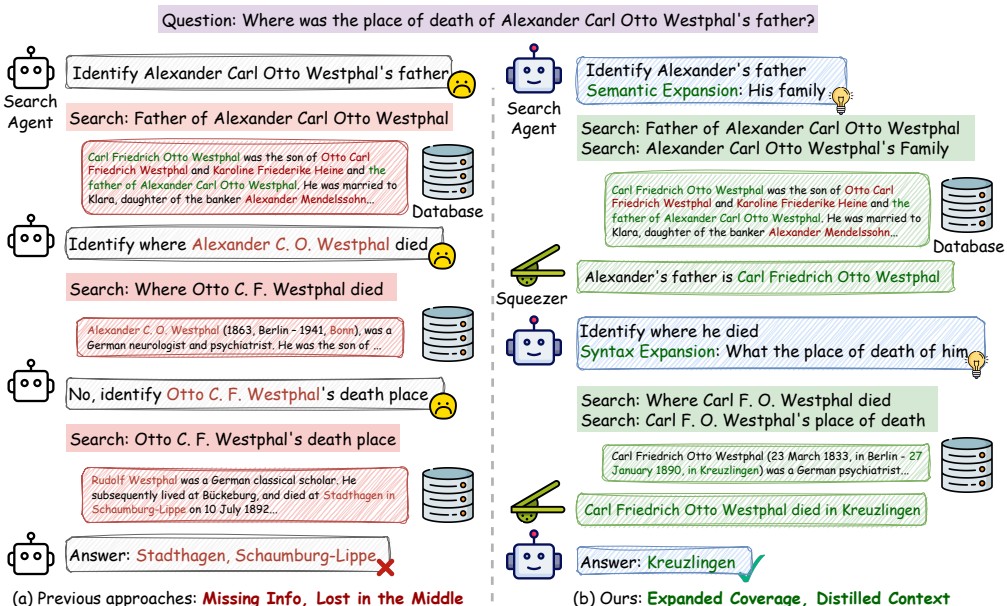

Figure 1: Comparison between Search-R1 and the proposed *ExpandSearch*. Our method generates multiple query variants to cover more information and achieve high retrieval recall. Meanwhile, we leverage a squeezer to compress the retrieved lengthy content into a compact passage, which preserves only the relevant information.

**Our key insight is that effective information retrieval requires a dual strategy: expanding the query space to maximize relevant information coverage, then selectively distilling the retrieved content to preserve only reasoning-critical facts.** This expand-then-squeeze paradigm mirrors human information-seeking behavior, where we cast a wide net through diverse search formulations, then critically filter and synthesize the gathered information. By explicitly separating the recall-optimization phase from the precision-optimization phase, we can address both semantic incompleteness and information overload systematically.

Therefore, we propose *ExpandSearch*, a novel reinforcement learning framework that trains search agents to perform query expansion while leveraging selective distillation for improved reasoning. In the expansion phase, we train models via reinforcement learning to reformulate initial queries into multiple semantically-enriched variants that capture diverse perspectives, alternative phrasings, and implicit semantic relationships. These expansions consist of two complementary types: syntax expansions that handle surface-form variations and semantic expansions that broaden the information scope, both addressing distinct retrieval limitations, as shown in Figure 1. This query diversification significantly improves retrieval recall, particularly for multi-hop reasoning tasks requiring evidence aggregation from multiple angles. In the squeeze phase, we prompt an LLM as a summarizer to perform selective distillation, extracting and condensing only the reasoning-relevant information from the expanded retrieval results while filtering out noise and redundancy. The search agent is trained to effectively utilize these distilled, reasoning-focused summaries to generate accurate answers.

The main contributions of our work are as follows:

- We identify and formalize the dual problems of semantic incompleteness and information overload in reasoning-augmented search agents, demonstrating through empirical analysis that both issues significantly degrade performance on complex reasoning tasks.

- We propose *ExpandSearch*, an expand-then-squeeze framework that combines reinforcement learning-based query expansion with prompted selective information distillation to achieve high recall while maintaining precision in multi-step reasoning scenarios.

- We demonstrate through extensive experiments across seven benchmarks that our method achieves substantial improvements over state-of-the-art baselines, with particularly strong gains on multi-hop reasoning tasks requiring diverse evidence aggregation.

## 2 RELATED WORK

### 2.1 DEEP SEARCH AGENTS

Large Language Models (LLMs) demonstrate remarkable reasoning capabilities (Achiam et al., 2023; Team et al., 2023; Dubey et al., 2024), yet remain fundamentally constrained by limited domain-specific expertise (Li et al., 2023) and susceptibility to hallucination phenomena (Ji et al., 2023). The integration of external knowledge sources through retrieval mechanisms has emerged as a critical solution, primarily manifesting in two paradigms: retrieval-augmented generation (RAG) (Guu et al., 2020) and the deployment of retrievers as interactive tools (Schick et al., 2023). RAG-based systems (Gao et al., 2023) typically implement a two-stage pipeline where retrieved documents are first obtained based on query similarity, then incorporated during the generation process. However, such approaches frequently struggle with the inclusion of contextually irrelevant information that can degrade output quality (Jin et al., 2025a).

The alternative search-as-a-tool framework empowers LLMs to actively invoke retrieval systems through either strategic prompting or targeted fine-tuning. Prompting-based approaches such as IR-CoT (Trivedi et al., 2023) and ReAct (Yao et al., 2023) guide models to interleave reasoning steps with strategic retrieval queries, while Toolformer (Schick et al., 2023) leverages supervised fine-tuning to enhance the model's search invocation capabilities. More recently, Search-R1 (Jin et al., 2025b) has pioneered the application of reinforcement learning techniques (Guo et al., 2025) to develop sophisticated search agents, demonstrating substantial performance improvements. Subsequent developments including ZeroSearch (Sun et al., 2025), MaskSearch (Wu et al., 2025), IKEA (Huang et al., 2025), HybridDeepSearcher (Ko et al., 2025), and ParallelSearch (Zhao et al., 2025) have introduced increasingly refined reward mechanisms to enhance search effectiveness. However, current approaches generate semantically limited queries that fail to capture diverse information aspects and struggle with information overload from irrelevant retrieved content. Our ExpandSearch framework addresses these limitations through an expand-then-squeeze paradigm that trains agents to generate semantically-enriched query expansions while leveraging selective distillation to extract reasoning-critical information from the retrieved results.

### 2.2 REINFORCEMENT LEARNING

Reinforcement learning (RL) enables agents to optimize cumulative rewards through iterative environmental interaction and feedback-driven learning (Sutton & Barto, 1998). The integration of RL with LLM training was pioneered by Ouyang et al. (2022) through reinforcement learning from human feedback (RLHF) (Kaufmann et al., 2023). RLHF employs human preference annotations (Lambert et al., 2025) to train reward models that guide LLM policy refinement, typically via Proximal Policy Optimization (PPO) (Schulman et al., 2017). However, PPO's requirement for repeated model updates creates significant computational bottlenecks. Recent efficiency-focused approaches including DPO (Rafailov et al., 2023), SimPO (Meng et al., 2024), and ORPO (Hong et al., 2024) circumvent reward model training entirely. Despite reduced computational costs, these methods face off-policy constraints (Pang et al., 2024) and typically yield inferior performance compared to traditional RL approaches. Emerging techniques like Group Relative Policy Optimization (GRPO) (Guo et al., 2025) and RLOO (Ahmadian et al., 2024) offer promising alternatives by eliminating critic networks through group-wise evaluation strategies. Recent work has demonstrated RL's effectiveness in improving LLM search and reasoning capabilities. Our framework leverages verifiable rewards to train agents for query expansion and effective utilization of distilled information, improving both semantic coverage and the processing of reasoning-focused summarization.

## 3 METHOD

In this section, we introduce the proposed expand-then-squeeze approach, the training template and the reward functions.

### 3.1 EXPAND-THEN-SQUEEZE STRATEGY

Similar to Search-R1, the proposed *ExpandSearch* adopts an iterative process. It alternates between the LLM generation and the search engine calling. Unlike Search-R1 which generates a single

---

**Algorithm 1** LLM Response Rollout with Multi-Turn Search Engine Calls in Parallel

---

**Require:** Input query $x$, policy model $\pi_\theta$, search engine $\mathcal{R}$, squeezer $\pi_s$, maximum turns $B$.
**Ensure:** Final response $y$.
1: Initialize rollout sequence $y \leftarrow \emptyset$
2: Initialize action count $b \leftarrow 0$
3: **while** $b < B$ **do**
4:     Initialize current action LLM rollout sequence $y_b \leftarrow \emptyset$
5:     **while** True **do**
6:         Generate response token $y_t \sim \pi_\theta(\cdot \mid x, y + y_b)$
7:         Append $y_t$ to rollout sequence $y_b \leftarrow y_b + y_t$
8:         **if** $y_t$ in [`</search>`, `</answer>`, `<eos>`] **then** break
9:         **end if**
10:     **end while**
11:     $y \leftarrow y + y_b$
12:     **if** `<search> </search>` detected in $y_b$ **then**
13:         Extract multiple queries $[q_1, q_2, \cdots, q_n] \leftarrow \text{Parse}(y_b, \text{<search>}, \text{</search>})$
14:         Retrieve search results in parallel $\{\mathcal{C}_i = \mathcal{R}(q_i)\}_{i=1}^n$
15:         Summarize the search results through squeezer $s = \pi_s(\{\mathcal{C}_i\}_{i=1}^n)$
16:         Insert the summary $s$ into rollout $y \leftarrow y + \text{<information>}s\text{</information>}$
17:     **else if** `<answer> </answer>` detected in $y_b$ **then**
18:         **return** final generated response $y$
19:     **else**
20:         Ask for rethink $y \leftarrow y+$ "My action is not correct. Let me rethink."
21:     **end if**
22:     Increment action count $b \leftarrow b + 1$
23: **end while**
24: **return** final generated response $y$

---

query in each step, ExpandSearch generates multiple diverse query variants, such as paraphrases, decomposed sub-questions, keyword expansions to facilitate retrieval for more relevant knowledge. We term this process as **expand**. It is worth noting that multiple expanded queries would retrieve a large number of chunks, bringing a huge amount of irrelevant information, significantly distracting the attention of LLM, especially in multi-turn rollout scenarios. To resolve this, we add a **squeeze** step. It compresses the retrieved long context into a compact text through a fixed-weight long-context LLM through API calls. We term the expansion followed by squeezing process as Expand-then-Squeeze strategy. In the following, we explain the proposed Expand and Squeeze strategy in detail.

**Expand.** If an external retrieval is needed, the LLM would generate a `<search> </search>` block wrapping up the set of n diverse queries $\{q_i\}_{i=1}^n$. For each query $q_i$, we call the search engine $\mathcal{R}$ to retrieve $k$ most relevant chunks:

$$\mathcal{C}_i = [c_i^1, \cdots, c_i^k] \leftarrow \mathcal{R}(q_i) \tag{1}$$

The expand step effectively overcomes the limitation of a single query (Weller et al., 2025) for retrieval, and improves the retrieval recall.

**Squeeze.** The generated queries $[q_1, \cdots, q_n]$ as well as the retrieved chunks $[\mathcal{C}_1, \cdots, \mathcal{C}_n]$ are fed into a frozen LLM with a prompt template defined in Section B, *e.g.,* squeezer, to squeeze the long retrieved chunks, $\pi_s$, to generate the summary:

$$s = \pi_s([q_1, \cdots, q_n], [\mathcal{C}_1, \cdots, \mathcal{C}_n]). \tag{2}$$

The squeezed information $s$ would be encapsulated in a block `<information> </information>` and inserted into the ongoing roll-out sequence, serving as an additional context for the next generation step. This search-then-reason iteration continues until the model generates a final response, which is enclosed between `<answer> </answer>`. The complete workflow is described in Algorithm 1. Compared to the original retrieved chunks $[\mathcal{C}_1, \cdots, \mathcal{C}_n]$,

Table 1: **Template for *ExpandSearch*.** question will be replaced with the specific question during training and inference.

---

Answer the given question. You must conduct reasoning inside `<think>` and `</think>` first every time you get new information. After reasoning, if you find you lack some knowledge, you can call a search engine by `<search>` query `</search>`, and it will return the searched results between `<information>` and `</information>`. Within `<search>` `</search>`, generate k diverse query variants — such as paraphrases, decomposed sub-questions, keyword expansions to facilitate retrieval for more relevant knowledge. Separate multiple queries with ## so they can be run in parallel.

Example format: `<search>` query_1 ## query_2 ## ... ## query_n `</search>`

You can search as many times as you want. If you find no further external knowledge needed, you can directly provide the answer inside `<answer>` and `</answer>` without detailed illustrations. For example, `<answer>` abc `</answer>`. Question: question.

---

the compressed information $s$ is much shorter, significantly saving the GPU memory consumption in the training instance. Meanwhile, as the weights of the squeezer are fixed in training, we host the squeezer independently in the inference instance, and the training instance calls the squeezer through an API.

## 3.2 TRAINING TEMPLATE AND REWARD FUNCTION

To train ExpandSearch, we create a simple template that instructs LLM to conduct reasoning-search iterations and generate the answer in the final step. As shown in Table 1, we encourage LLM to explicitly conduct query expansion and generate $k$ various query variants to achieve high recall.

Following Search-R1, the reward function for training ExpandSearch is a weighted sum of the exact-match (EM) reward $r_{\mathrm{EM}}$ and the format reward $r_{\mathrm{f}}$:

$$r = r_{\mathrm{EM}} + \lambda r_{\mathrm{f}}. \tag{3}$$

The extract-match (EM) reward is defined as $r_{\mathrm{EM}}(\mathrm{ans}_{\mathrm{pred}}, \mathrm{ans}_{\mathrm{gt}}) = \mathbb{I}(\mathrm{ans}_{\mathrm{pred}} = \mathrm{ans}_{\mathrm{gt}})$ where $\mathbb{I}(\cdot)$ denotes the indicator function, which is 1 only for the condition that the predicted answer $\mathrm{ans}_{\mathrm{pred}}$ is exactly the same as the ground-truth answer $\mathrm{ans}_{\mathrm{gt}}$. The format reward $r_{\mathrm{f}}$ is 1 only if the predicted answer $\mathrm{ans}_{\mathrm{pred}}$ strictly follows the format defined in Table 1. By default, we set $\lambda$ as 0.2.

## 4 RESULTS AND ANALYSIS

### 4.1 EXPERIMENT SETUP

**Datasets and Evaluation Metric** The evaluation includes seven benchmarks spanning diverse retrieval complexities, grouped into two categories: (1) General Question Answering, comprising NQ (Kwiatkowski et al., 2019), TriviaQA (Joshi et al., 2017), and PopQA (Mallen et al., 2023); (2) Multi-Hop Question Answering, including HotpotQA (Yang et al., 2018), 2WikiMultiHopQA (Ho et al., 2020), Musique (Trivedi et al., 2022), and Bamboogle (Press et al., 2023). Following Jin et al. (2025b), we combine NQ and HotpotQA training sets for training, evaluating on validation/test splits using Exact Match (EM) to measure both in-domain and out-of-domain generalization.

**Baseline Methods** We employ various baselines to evaluate ExpandSearch, including R1 without search engine (Guo et al., 2025), Search-R1 (Jin et al., 2025b), ZeroSearch (Sun et al., 2025), StepSearch (Wang et al., 2025), Route-R1 Zhang et al. (2025) and ParallelSearch (Zhao et al., 2025).

**Implementation Details** We conduct experiments using Qwen-2.5-Base/Instruct models (Yang et al., 2024) as the backbone of the search agent, E5 (Wang et al., 2022) as the embedding model, and the 2018 Wikipedia dump (Karpukhin et al., 2020) as the corpus. All experiments are conducted on 8 NVIDIA H100 GPUs. The model is developed with the Search-R1 framework and trained via veRL for reinforcement learning, using Proximal Policy Optimization (PPO) as the default algorithm. More details can be found in Section A.

Table 2: **Exact Match (EM) scores across seven general and multi-hop question answering benchmarks. Bold** indicates the best performance. $^{\dagger}/^{\star}$ denote in-domain/out-of-domain datasets. $^{\ddagger}$ represents methods trained on different training sets. *ExpandSearch* consistently outperforms baselines in average performance.

| Methods | General QA | | | Multi-Hop QA | | | | Avg. |
|---|---|---|---|---|---|---|---|---|
| | NQ$^{\dagger}$ | TriviaQA$^{\star}$ | PopQA$^{\star}$ | HotpotQA$^{\dagger}$ | 2wiki$^{\star}$ | Musique$^{\star}$ | Bamboogle$^{\star}$ | |
| *Qwen2.5-3b-Instruct* | | | | | | | | |
| R1 | 0.210 | 0.449 | 0.171 | 0.208 | 0.275 | 0.060 | 0.192 | 0.224 |
| Search-R1 | 0.341 | 0.545 | 0.378 | 0.324 | 0.319 | 0.103 | 0.264 | 0.325 |
| ZeroSearch$^{\ddagger}$ | 0.414 | 0.574 | 0.448 | 0.274 | 0.300 | 0.098 | 0.111 | 0.317 |
| StepSearch$^{\ddagger}$ | - | - | - | 0.345 | 0.320 | 0.174 | 0.344 | - |
| Router-R1 | 0.388 | **0.706** | 0.384 | 0.352 | 0.434 | 0.138 | 0.512 | 0.416 |
| *ExpandSearch* (*Ours*) | **0.446** | 0.677 | **0.456** | **0.422** | **0.450** | **0.194** | **0.540** | **0.457** |
| *Qwen2.5-3b-Base* | | | | | | | | |
| R1 | 0.226 | 0.455 | 0.173 | 0.201 | 0.268 | 0.055 | 0.224 | 0.229 |
| Search-R1 | 0.406 | 0.587 | 0.435 | 0.284 | 0.273 | 0.049 | 0.088 | 0.303 |
| ZeroSearch$^{\ddagger}$ | 0.430 | 0.616 | 0.414 | 0.338 | 0.346 | 0.130 | 0.139 | 0.345 |
| StepSearch$^{\ddagger}$ | - | - | - | 0.329 | 0.339 | **0.181** | 0.328 | - |
| *ExpandSearch* (*Ours*) | **0.488** | **0.700** | **0.507** | **0.414** | **0.398** | 0.136 | **0.452** | **0.435** |
| *Qwen2.5-7b-Instruct* | | | | | | | | |
| R1 | 0.270 | 0.537 | 0.199 | 0.237 | 0.292 | 0.072 | 0.293 | 0.271 |
| Search-R1 | 0.393 | 0.610 | 0.397 | 0.370 | 0.414 | 0.146 | 0.368 | 0.385 |
| ZeroSearch$^{\ddagger}$ | 0.436 | 0.652 | **0.488** | 0.346 | 0.352 | 0.184 | 0.278 | 0.391 |
| StepSearch$^{\ddagger}$ | - | - | - | 0.386 | 0.366 | **0.226** | 0.400 | - |
| ParallelSearch | **0.462** | 0.628 | 0.429 | **0.429** | 0.424 | 0.197 | 0.411 | 0.425 |
| *ExpandSearch* (*Ours*) | 0.450 | **0.667** | 0.451 | 0.428 | **0.459** | 0.211 | **0.476** | **0.449** |
| *Qwen2.5-7b-Base* | | | | | | | | |
| R1 | 0.297 | 0.539 | 0.202 | 0.242 | 0.273 | 0.083 | 0.296 | 0.276 |
| Search-R1 | 0.480 | 0.638 | 0.457 | 0.433 | 0.382 | 0.196 | 0.432 | 0.431 |
| ZeroSearch$^{\ddagger}$ | 0.424 | 0.664 | **0.604** | 0.320 | 0.340 | 0.180 | 0.333 | 0.409 |
| StepSearch$^{\ddagger}$ | - | - | - | 0.380 | 0.385 | 0.216 | 0.467 | - |
| ParallelSearch | 0.492 | 0.658 | 0.455 | **0.457** | 0.452 | **0.229** | 0.468 | 0.458 |
| *ExpandSearch* (*Ours*) | **0.496** | **0.703** | 0.506 | 0.445 | **0.488** | 0.196 | **0.540** | **0.480** |

## 4.2 MAIN RESULTS

Table 2 presents the results comparing *ExpandSearch* against baselines across seven question-answering benchmarks and four model configurations. Our analysis reveals several key findings.

**(1) Our method achieves consistent and substantial improvements across all configurations.** It outperforms the strongest baselines by an average of $4.4\%$ absolute improvement. This demonstrates the robustness and effectiveness of our reinforcement learning approach for search agent training.

**(2) 3B search agents can surpass larger 7B counterparts.** Remarkably, our 3B-Instruct model achieves $0.457$ average EM score, outperforming 7B baseline methods, including Search-R1 and ZeroSearch by substantial margins, and is comparable with the SOTA 7B model ParallelSearch, showing that the expand stage increases the retrieval coverage and the squeeze stage provides the most relevant information to search agents.

**(3) Model architecture and instruction tuning interact differently across scales.** For 3B models, the instruct variant outperforms the base model by $2.2\%$, because smaller base models struggle with instruction following and cannot generate correct search operations during the RL rollout stage to get a valid trajectory. Conversely, for 7B models, the base variant surpasses the instruct model by $3.1\%$, suggesting that at larger scales, base models retain more flexible capabilities from pre-training that benefit search agent behavior when both variants can adequately follow instructions.

**(4) Our method consistently improves the performance on both general and multi-hop tasks.** Our method achieves an average improvement of $5.0\%$ and $4.0\%$ over the best baseline on the general and multi-hop QA benchmarks, respectively, indicating the agent effectively learns to decompose complex queries and orchestrate multi-step search strategies.

**(5) Performance gains are consistent across in-domain and out-of-domain evaluations.** Our method shows robust generalization with $5.2\%$ average improvement on out-of-domain benchmarks and $3.0\%$ on in-domain benchmarks, suggesting that the learned search policies transfer well to unseen question distributions.

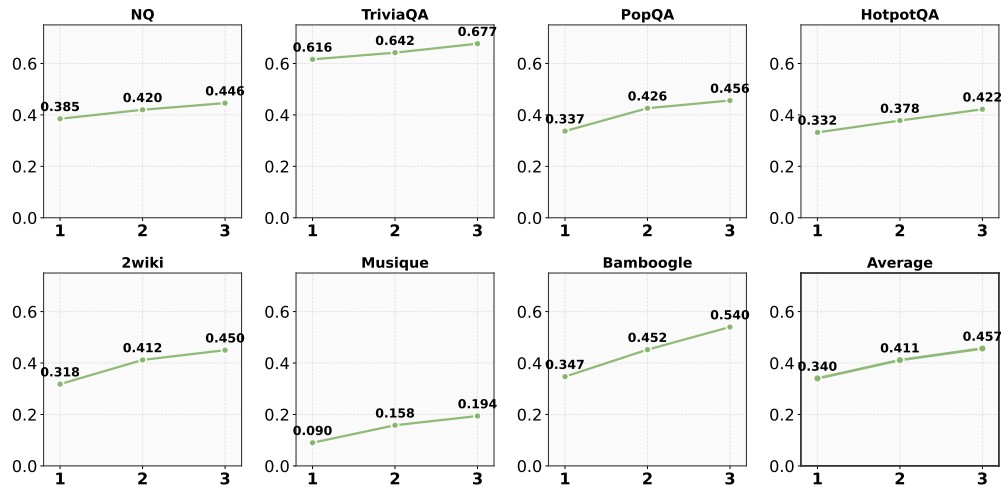

Figure 2: The influence of the number of rephrased queries, $n$, on EM accuracy.

Table 3: Ablation study comparing trained query expansion with untrained expansion and the impact of the squeezer component.

| Methods | General QA | | | Multi-Hop QA | | | | Avg. |
|---|---|---|---|---|---|---|---|---|
| | NQ | TriviaQA | PopQA | HotpotQA | 2wiki | Musique | Bamboogle | |
| *ExpandSearch* | **0.444** | **0.664** | **0.447** | **0.415** | **0.432** | **0.196** | **0.524** | **0.446** |
| - w/o squeezer | 0.385 | 0.570 | 0.398 | 0.361 | 0.384 | 0.146 | 0.323 | 0.364 |
| Search-R1 | 0.352 | 0.557 | 0.395 | 0.324 | 0.321 | 0.111 | 0.266 | 0.332 |
| - w/ Expansion + Squeezer | 0.335 | 0.543 | 0.340 | 0.327 | 0.303 | 0.099 | 0.360 | 0.330 |

## 4.3 EXPANSION BEHAVIOR ANALYSIS

**Query expansion significantly improves retrieval recall by addressing the semantic brittleness of single-query search.** As demonstrated in Figure 2, increasing the number of expanded queries from 1 to 3 yields consistent improvements in exact match accuracy across benchmarks. The most substantial gains occur when moving from single-query ($n = 1$) to dual-query ($n = 2$) generation, with an average improvement of $6.7\%$, suggesting that even minimal expansion captures previously missed semantic variations. The continued improvement at $n = 3$ indicates that complex reasoning tasks benefit from multiple complementary query perspectives, though with diminishing returns that suggest a natural saturation point in query diversity.

**End-to-end reinforcement learning is essential for learning effective query expansion strategies.** We compare our method with $n$ set to 3 and $k$ set to 5 to Search-R1 with $k = 15$, i.e., both methods have the same number of retrieval chunks. As shown in Table 3, our results demonstrate the critical importance of end-to-end training for query expansion. ExpandSearch achieves an average EM score of $0.446$, representing a $34.3\%$ relative improvement over Search-R1. This substantial gain validates our hypothesis that trained query expansion can effectively address the semantic limitations of single-query retrieval. A particularly revealing finding is that simply adding the expansion prompt in Table 1 and the squeezer to Search-R1 without RL training (`Search-R1 w/ Expansion + Squeezer`) yields no improvement and even slightly degrades performance. This counterintuitive result highlights a crucial insight that naive query expansion without proper training can introduce noise that overwhelms the potential benefits of increased coverage. In contrast, when we remove the squeezer and only train the agent using a query expansion (`ExpandSearch w/o squeezer`) outperforms the baselines, showing that (1) generating semantically coherent and diverse query variants that capture different aspects of the information need, and (2) properly utilizing the expanded retrieval results in the reasoning process. Without end-to-end training, the model cannot learn the complex interplay between query generation and answer generation.

Table 4: Impact of removing syntax and semantic expansions on EM accuracy.

| Variants | Ratio | General QA | | | Multi-Hop QA | | | | Avg. |
|---|---|---|---|---|---|---|---|---|---|
| | | NQ | TriviaQA | PopQA | HotpotQA | 2wiki | Musique | Bamboogle | |
| *ExpandSearch* | - | **0.444** | **0.664** | **0.447** | **0.415** | **0.432** | **0.196** | **0.524** | **0.446** |
| w/o Syntax Expansion | 63.35% | 0.443 | 0.636 | 0.381 | 0.404 | 0.407 | 0.191 | 0.384 | 0.407 |
| w/o Semantic Expansion | 36.65% | 0.438 | 0.644 | 0.370 | 0.413 | 0.402 | 0.175 | 0.368 | 0.401 |

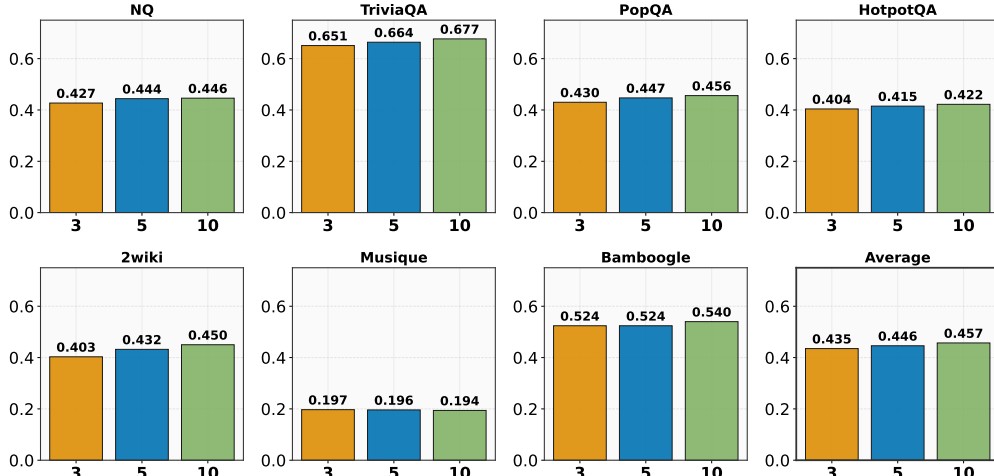

Figure 3: The influence of the number of retrieved chunks, $k$, on EM accuracy.

**Our learned query expansion strategy naturally discovers two complementary expansion types that address distinct retrieval failure modes.** Through qualitative analysis in Section E of generated queries, we observe that: (1) Syntactic expansions handle surface-form variations that confuse embedding models. For instance, expanding "where did he die" to include "his death place" and "location of death" overcomes word-order sensitivity in dense retrievers; (2) Semantic expansions broaden the information scope by generating related but distinct queries, such as expanding "Alex's father" to include "Alex's family" and "Alex's parents", capturing information that may be stored under broader categorical terms.

**Syntax expansions dominate the learned query distribution, reflecting the brittleness of embedding-based retrieval to surface forms.** We use an LLM to analyze and assign a type of each query. As shown in Table 4, our trained model generates 63.35% syntax expansions versus 36.65% semantic expansions, suggesting that paraphrasing and reformulation are the primary mechanisms for improving retrieval recall. This distribution emerges naturally through reinforcement learning, indicating that lexical variation poses a greater challenge than semantic breadth for current dense retrievers. More details can be found in Section B.

**Both expansion types are essential and non-redundant for optimal performance.** Removing syntax or semantic expansions leads to consistent performance decline, demonstrating that neither expansion type can fully compensate for the other. This complementarity suggests they address fundamentally different retrieval limitations.

### 4.4 SQUEEZE BEHAVIOR ANALYSIS

**The squeezer component is critical for managing information overload from expanded retrieval.** As shown in Table 3, removing the squeezer from ExpandSearch causes a dramatic performance drop. This degradation is particularly severe on complex multi-hop tasks like Bamboogle and Musique, where the model must navigate through multiple chunks of potentially irrelevant information without selective distillation.

**Retrieval depth shows consistent but diminishing returns across all benchmarks.** Figure 3 demonstrates that increasing $k$ from 3 to 5 chunks yields substantial improvements, while further

Table 5: Impact of squeezer selection on EM accuracy.

| Squeezer | General QA | | | Multi-Hop QA | | | | Avg. |
|---|---|---|---|---|---|---|---|---|
| | NQ | TriviaQA | PopQA | HotpotQA | 2wiki | Musique | Bamboogle | |
| LLaMA-3.1-8b | 0.411 | 0.611 | 0.425 | 0.386 | 0.377 | 0.159 | 0.379 | 0.389 |
| LLaMA-3.1-70b | **0.481** | **0.682** | **0.476** | 0.404 | 0.392 | 0.167 | 0.476 | 0.433 |
| LLaMA-4-17b | 0.446 | 0.677 | 0.456 | **0.422** | **0.450** | **0.194** | **0.540** | **0.457** |

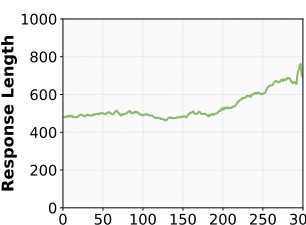 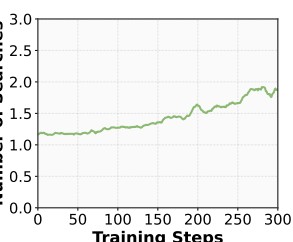

Figure 4: Training dynamics during reinforcement learning.

expansion to $k = 10$ provides marginal benefits for some benchmarks. This saturation pattern validates our squeeze-based approach rather than unlimited retrieval expansion, selective distillation of moderately-sized retrieval sets proves more effective.

**Squeezer architecture exhibits a task-specific trade-off between model capacity and specialization.** As shown in Table 5, LLaMA-3.1-70B achieves superior performance on general QA tasks, leveraging its larger capacity to better understand diverse queries. However, LLaMA-4-17b, despite being smaller, outperforms on multi-hop reasoning tasks. It reveals that general knowledge extraction and multi-hop evidence synthesis require fundamentally different compression capabilities.

Results regarding testing different squeezer models during inference time can be found in Section C.

### 4.5 TRAINING DYNAMICS

We illustrate the training dynamics in Figure 4. The reinforcement learning process effectively discovers the correlation between query expansion and task performance, as evidenced by the steady increase in reward. Notably, the model autonomously learns to increase search frequency as a strategy for improving answer quality, demonstrating that expanded retrieval emerges as an optimal strategy without explicit supervision. This behavioral shift correlates with response length expansion, suggesting that the model learns to synthesize information from multiple retrieval rounds into more comprehensive answers. The synchronized growth of all three metrics reveals a coherent learned strategy where more searches yield more relevant evidence, enabling more detailed and accurate responses. The smooth training curves without sudden drops or high variance indicate stable optimization, suggesting that the expand-then-squeeze framework provides a robust learning signal that allows progressive refinement of search strategies while maintaining previously learned capabilities.

### 5 CONCLUSION

In this paper, we presented *ExpandSearch*, a reinforcement learning framework that trains search agents to overcome the limitations of single-query retrieval through learned query expansion and selective information distillation. Our expand-then-squeeze paradigm addresses two critical challenges in reasoning-augmented search: semantic incompleteness in query formulation and information overload from expanded retrieval sets. Experiments across seven QA benchmarks demonstrate that ExpandSearch achieves substantial improvements over state-of-the-art baselines, with particularly strong gains on multi-hop reasoning tasks. These results suggest that the future of retrieval-augmented reasoning lies not in simply scaling retrieval volume or model size, but in learning sophisticated strategies for query formulation and information synthesis, opening avenues for further research into adaptive retrieval strategies and more efficient training methods for search agents.

## 6 ETHICS STATEMENT

This work adheres to the ICLR Code of Ethics. Our research uses only publicly available benchmark datasets (NQ, TriviaQA, PopQA, HotpotQA, 2WikiMultiHopQA, Musique, Bamboogle) with no human subjects involved. Our primary intent is to advance scientific understanding of retrieval-augmented reasoning.

## 7 REPRODUCIBILITY STATEMENT

To ensure reproducibility, we provide detailed implementation specifications throughout the paper and appendix. All experiments use publicly available models (Qwen-2.5-Base/Instruct 3B/7B variants) and standard benchmarks with established train/validation/test splits. Key hyperparameters including PPO algorithm settings, reward function weights, and training configurations are specified in Section 4.1. The E5 embedding model and 2018 Wikipedia dump used for retrieval are publicly accessible. Training is conducted using the veRL framework for reinforcement learning. The squeezer prompt template is provided in Table 6. We report exact match scores averaged across multiple model configurations. The algorithmic details in Algorithm 1 and training dynamics in Figure 4 provide sufficient information for reproduction.

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

# A  EXPERIMENT SETUP

## A.1  DATASETS

**Natural Questions (NQ)**: A large-scale open-domain QA dataset containing 307K training, 8K development, and 8K test examples derived from Wikipedia pages. The dataset supports both long answer selection (identifying relevant passages) and short answer extraction (finding specific answer spans), with human performance benchmarks of 87% F1 and 76% F1 respectively. NQ provides real questions posed by information-seeking users, making it particularly valuable for developing systems that address genuine information needs.

**TriviaQA**: A reading comprehension dataset with over 65K question-answer-evidence triples, featuring 95K QA pairs authored by trivia enthusiasts paired with independently gathered evidence documents (averaging six per question). The dataset challenges systems with compositional questions requiring complex understanding, significant lexical variability between questions and evidence, and cross-sentence reasoning requirements.

**PopQA**: A 14K entity-centric QA dataset designed to evaluate language models' factual knowledge across varying entity popularity levels. Questions are systematically generated from Wikidata tuples covering 16 relationship types, with each question annotated with entity popularity metrics from Wikipedia page views. The dataset uniquely emphasizes long-tail entities to assess how well models encode less popular factual information, revealing that while model scaling improves popular knowledge memorization, retrieval augmentation remains necessary for long-tail facts.

**HotpotQA**: A 113K Wikipedia-based QA dataset requiring genuine multi-hop reasoning across multiple documents. The dataset features diverse, unconstrained questions with sentence-level supporting fact annotations that enable explainable predictions and strong supervision. It includes a novel category of factoid comparison questions requiring fact extraction and comparative analysis, advancing both reasoning capabilities and interpretability in QA systems.

**2WikiMultiHopQA**: A multi-hop QA dataset with $192,606$ examples combining structured Wikidata and unstructured Wikipedia text to guarantee genuine multi-hop reasoning. The dataset features four question types (comparison, inference, compositional, and bridge comparison) with complete reasoning path annotations from question to answer. Answer types are diverse, including yes/no, dates, and $708$ unique answer categories total, ensuring broad coverage of reasoning patterns.

**MuSiQue**: A reading comprehension dataset addressing shortcut-based reasoning through bottom-up construction of $2-4$ hop questions from composed single-hop questions. Available in two variants—MuSiQue-Answerable (25K questions) and MuSiQue-Full (50K questions including unanswerable pairs)—the dataset requires connected reasoning where each step depends on previous information.

**Bamboogle**: A manually curated dataset of 125 carefully crafted 2-hop questions created by human annotators connecting unrelated facts from Wikipedia articles. Questions are quality-controlled by filtering out those answerable through simple web searches, ensuring genuine multi-hop reasoning requirements. Despite its smaller scale, Bamboogle provides valuable evaluation for authentic complex question decomposition beyond template-based patterns, complementing larger automatically generated datasets.

## A.2  BASELINES

We evaluate our proposed method against several state-of-the-art baselines: R1 without search engine, Search-R1, ZeroSearch, StepSearch, Router-R1, and ParallelSearch.

**R1 without search engine**: This baseline directly prompts DeepSeek-R1-Distill-Qwen to generate answers without accessing external data sources, serving as a non-retrieval baseline that relies solely on the model's parametric knowledge.

**Search-R1**: A reinforcement learning framework that trains LLMs to autonomously interleave reasoning with search engine interactions. The model learns to generate special tokens that trigger search queries during step-by-step reasoning. Using PPO or GRPO algorithms with outcome-based rewards, Search-R1 employs retrieved token masking to prevent optimization over external content

and develops dynamic search strategies through trial-and-error learning, including capabilities for self-verification and iterative refinement.

**ZeroSearch**: A reinforcement learning approach that trains LLMs to develop search capabilities without real search engines by using a lightweight fine-tuned LLM as a simulated search engine. The framework employs curriculum-based rollout strategies with progressively degraded document quality to expose the policy model to increasingly challenging retrieval scenarios. This method addresses the unpredictability of document quality and high API costs associated with real search engines while demonstrating that even 3B parameter simulators can effectively train search capabilities comparable to real search engine performance.

**StepSearch**: A reinforcement learning framework implementing step-wise proximal policy optimization with fine-grained, token-level supervision. The method enforces a structured think-search-answer loop and computes rewards based on information gain (measured via cosine similarity over TF-IDF representations) while penalizing redundant retrievals. This step-wise supervision enables effective decomposition of complex multi-hop queries into focused search subtasks with dynamic retrieval strategy adaptation.

**Router-R1**: A reinforcement learning-based multi-LLM routing framework that formulates model selection and aggregation as a sequential decision process. The router, instantiated as an LLM itself, interleaves "think" actions (internal deliberation) with "route" actions (dynamic model invocation) while integrating responses into evolving context. The framework employs lightweight rule-based rewards combining format rewards, outcome rewards, and cost optimization, conditioning on simple model descriptors (pricing, latency, example performance) to achieve strong generalization to unseen model selection scenarios.

**ParallelSearch**: A reinforcement learning framework that enables LLMs to recognize parallelizable query structures and execute multiple search operations concurrently, addressing the sequential bottleneck in existing search agents. The approach introduces specialized reward functions that jointly consider correctness, query decomposition quality, and parallel execution benefits.

## A.3 IMPLEMENTATION DETAILS

Our experimental framework employs Qwen-2.5 models (both Base and Instruct variants) as the search agent backbone, with E5 serving as the embedding model for retrieval tasks. For corpus construction, we utilize benchmark-specific data for MultihopRAG, while employing the 2018 Wikipedia dump for all other benchmarks. We use the retrieval configuration to 10 passages in this work. All experiments are performed using 8 NVIDIA H100 GPUs.

**Training Configuration**: The PPO algorithm is configured with differentiated learning rates: 1e-6 for the policy LLM and 1e-5 for the value LLM. Training proceeds for 500 steps with warm-up ratios set at 28.5% and 1.5% for policy and value models respectively. Generalized Advantage Estimation (GAE) parameters are configured with $\lambda = 1$ and $\gamma = 1$.

**Batch Processing**: We implement a hierarchical batch structure with a total batch size of 512, divided into mini-batches of 256 and micro-batches of 64. The maximum sequence length is constrained to $4,096$ tokens, with response generation limited to 500 tokens and retrieved content truncated at 500 tokens.

**System Optimization**: Memory efficiency is achieved through gradient checkpointing and Fully Sharded Data Parallel (FSDP) with CPU offloading. For inference, we deploy vLLM with 1 tensor parallelism, GPU memory utilization capped at 60%, and sampling parameters of 1.0 temperature and 1.0 top-p during rollout generation.

**Regularization Parameters**: The framework incorporates KL divergence regularization with coefficient $\beta = 0.001$ and PPO clipping with ratio $\epsilon = 0.2$ to ensure stable training dynamics.

## B PROMPT TEMPLATES

The prompt template used in squeezers is shown in Table 6. We also present the prompt template that used to analyze the expansion type in Table 4, as illustrated in Table 7.

Table 6: Prompt template for squeezers.

You are a helpful assistant.
You are given a series of queries and contexts.
Return the answer to queries based on the Contexts and nothing else.
Queries: QUERIES
Contexts: CONTEXT
Answer:

Table 7: Prompt template for analyzing expansion types.

Classify the following query expansion type.

Base Query: BASE_QUERY
Expanded Query: EXPANDED QUERY

Query expansion types:
- Syntax Expansion: Reformulating the query structure while keeping the same meaning (e.g., "Alexander's father" → "father of Alexander", "where did he die" → "death place of")
- Semantic Expansion: Expanding the meaning to related concepts (e.g., "Alexander's father" → "Alexander's family", "death place" → "burial location")

Respond with ONLY one word: 'syntax' or 'semantic'

## C  TEST-TIME GENERALIZATION ACROSS SQUEEZERS

**The expand-then-squeeze framework exhibits robust generalization across different squeezer models without requiring retraining.** As shown in Figure 5, when we replace the training-time squeezer (LLaMA-4-17b) with LLaMA-3.1-8b during inference, the system maintains comparable performance across all benchmarks, demonstrating that the search agent's learned expansion strategies are not tightly coupled to a specific squeezer implementation. This plug-and-play compatibility suggests that the agent learns generalizable query expansion patterns rather than squeezer-specific adaptations.

**Different squeezer architectures exhibit task-specific advantages despite similar overall performance.** Interestingly, the smaller LLaMA-3.1-8b actually outperforms the larger LLaMA-4-17b on certain benchmarks like NQ and PopQA, while the larger model excels on complex multi-hop tasks like Bamboogle and 2WikiMultiHopQA. This pattern reveals that squeezer selection can be optimized post-training based on task requirements—simpler factual queries benefit from faster, lighter squeezers while complex reasoning chains leverage the superior synthesis capabilities of larger models.

**The modular architecture enables flexible deployment strategies without performance degradation.** The consistent performance across different squeezers validates our architectural decision to decouple the expansion and squeeze components. This modularity allows practitioners to adjust the squeezer based on computational constraints or task-specific requirements without retraining the entire system. For instance, resource-constrained deployments could use LLaMA-3.1-8b with minimal performance loss, while accuracy-critical applications could employ larger squeezers for marginal gains on complex reasoning tasks.

## D  LLM USAGE STATEMENT

Large language models were used as a general-purpose writing assistance tool during the preparation of this manuscript, primarily for grammar checking, sentence restructuring, and improving clarity of technical descriptions. LLMs did not contribute to the core research ideas, experimental design, or technical innovations presented in this work. All scientific claims, experimental results, and theoretical contributions are the original work of the authors, who take full responsibility for the accuracy and integrity of all content.

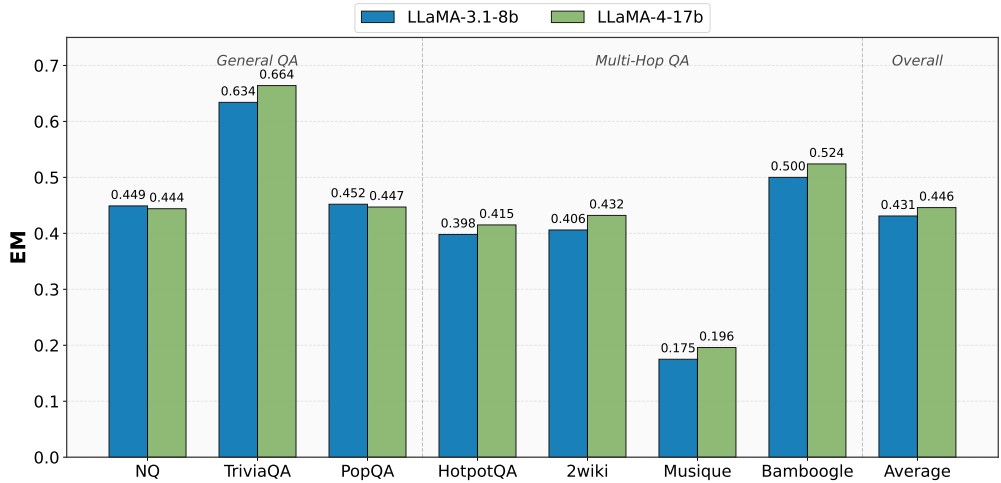

Figure 5: Performance comparison of ExpandSearch using different squeezer models at inference time.

Table 8: Case study using the question from HotpotQA.

| |
|---|
| **Question**: 2014 S/S is the debut album of a South Korean boy group that was formed by who? |
| **Ground Truth**: YG Entertainment |
| *ExpandSearch*: |
| `<think>` To determine who formed the boy group, I need to first identify the boy group. `</think>` |
| `<search>` boy group that debuted with the album 2014 S/S ## male group first album 2014 S/S debut ## K-pop boy groups debut albums 2014 rookie artists `</search>` |
| `<information>` Doc 1 Doc 2 ... Based on the provided contexts, here are the answers to the given queries: WINNER is the boy group. `</information>` |
| `<think>` Now that I know the boy group that debuted with the album 2014 S/S is WINNER. I can directly find who formed them. `</think>` |
| `<search>` who formed the boy group WINNER ## WINNER boy group created by who ## who created the WINNER `</search>` |
| `<information>` Doc 1 Doc 2 ... Based on the provided contexts, here are the answers to the given queries: YG Entertainment. `</information>` |
| `<answer>` YG Entertainment `</answer>` |

# E  CASE STUDIES

We conduct case studies to gain a deeper understanding of the behavior and capabilities of our *ExpandSearch*. The results are shown in the following tables.

Table 9: Case study using the question from 2wiki.

| |
|---|
| **Question**: When did John V, Prince Of Anhalt-Zerbst's father die? |
| **Ground Truth**: 12 June 1516 |
| *ExpandSearch*: |

`<think>` To determine when John V, Prince of Anhalt-Zerbst's father died, I need to first identify who his father was. `</think>`
`<search>` father of John V, Prince of Anhalt-Zerbst ## John V Prince of Anhalt-Zerbst's father ## Anhalt-Zerbst royal family tree German princes 17th century genealogy `</search>`
`<information>` Doc 1 Doc2 ... Based on the provided contexts, here are the answers to the given queries: Ernest I, Prince of Anhalt-Dessau. `</information>`
`<think>` Now that I know his father. I can directly find when he died. `</think>`
`<search>` Ernest I Prince of Anhalt-Dessau death date ## when did Ernest I, Prince of Anhalt-Dessau die ## Anhalt-Dessau rulers 16th century German princes death dates biography `</search>`
`<information>` Doc 1 Doc 2 ... Based on the provided contexts, here are the answers to the given queries: 12 June 1516. `</information>`
`<answer>` 12 June 1516 `</answer>`

