# OpenReview forum: "Beyond the limitation of a single query: Train your LLM for query expansion with Reinforcement Learning"
_ICLR.cc/2026/Conference — Submitted to ICLR 2026_

### Official Review · Reviewer_Mg5C · 2025-10-29

**Soundness:** 3
**Presentation:** 3
**Contribution:** 3
**Rating:** 6
**Confidence:** 4

**Summary:**

The paper proposes ExpandSearch, a search agent trained with reinforcement learning to generate multiple complementary query variants (syntax-/semantics-oriented) and then compress retrieved chunks with a frozen “squeezer” LLM before continuing reasoning. Across seven QA benchmarks (NQ, TriviaQA, PopQA, HotpotQA, 2Wiki, MuSiQue, Bamboogle), ExpandSearch outperforms baselines and achieves an average EM of 0.446; ablations show a 34.3% relative gain over Search-R1 and a clear drop without the squeezer, while naïvely adding “expansion+squeeze” without RL yields no improvement.

**Strengths:**

1. End-to-end “expand-then-squeeze” design directly targets single-query semantic brittleness and information overload: parallel query expansion boosts recall, a frozen squeezer condenses evidence, and the agent is trained with PPO within the Search-R1 framework.

2. Broad, consistent gains on seven benchmarks with clear reporting of training/eval splits and baselines; average EM 0.446 and a 34.3% relative improvement over Search-R1 substantiate the claim.

3. Careful experiment design: EM improves as the number of rephrased queries n rises (with diminishing returns), ablations quantify the squeezer’s contribution (avg EM 0.446 → 0.364 when removed), and test-time generalization holds across different squeezers (LLaMA-4-17b vs LLaMA-3.1-8b).

**Weaknesses:**

1. Efficiency and cost are insufficiently characterized given the added query fan-out and extra squeezer calls; figures analyze accuracy vs. n but omit end-to-end latency, token/compute costs, or cost-accuracy trade-off curves.

2. The squeezer is frozen and not jointly trained with the agent, preventing cross-module credit assignment and co-adaptation; joint or partially joint training is absent and could unlock further gains.

3. No out-of-domain evaluation: models trained/evaluated on Wikipedia are not tested under corpus shift (e.g., biomedical, legal, finance); assessing OOD generalization on domains like PubMed/BioASQ (or comparable corpora) with retrieval quality, answer accuracy, and calibration would strengthen external validity.

**Questions:**

refer to weaknesses

---

> ### Author Response · Authors · 2025-11-27
>
> We sincerely thank the reviewer for the thoughtful and constructive feedback. We greatly appreciate your recognition of "_careful experiment design_" and "_strong results_." We address each of your valuable questions in detail below.
>
> ---
>
> > ***`Q1`: Computational Efficiency and Latency***
>
> `A`: We appreciate this concern regarding practical deployment considerations. **Counterintuitively, ExpandSearch achieves lower latency than baselines.**
>
> * 3B Baseline. The smaller model struggles to gather sufficient information, resulting in repeated query generation across multiple rounds, significantly increasing overall latency. This phenomenon is also documented in recent work such as [1].
> * 7B Baseline. While more capable, the larger model incurs higher per-token inference costs.
>
> **ExpandSearch's decoupled architecture** assigns the search task to an efficient 3B model while delegating extraction to a specialized squeezer, optimizing the compute-capability tradeoff.
>
> **Latency Comparison (ms per sample):**
>
> | Configuration                          | Latency (ms) |
> | -------------------------------------- | ------------ |
> | Search-R1 3B                           | 59.91        |
> | Search-R1 7B                           | 31.12        |
> | ExpandSearch (3B + Llama3-8B Squeezer) | **23.47**    |
>
>
> ---
>
> > ***`Q2`: Frozen Squeezer***
>
> `A`: In this work, we focus on demonstrating the effectiveness of the Expand-then-Squeeze pipeline. Additionally, the frozen squeezer enables **plug-and-play selection** of squeezer sizes (`Figure 5, Appendix`), potentially establishing a new paradigm where routers dynamically schedule squeezer sizes based on query difficulty without retraining.
>
> Unlocking and training the squeezer would escalate the task to **multi-agent RL training**, which is a distinct but important research area. We consider this a promising future direction and will discuss it in the revised manuscript.
>
> ---
>
> > ***`Q3`: Cross-Domain Generalization***
>
> `A`: Thanks for your question. Following s3, we evaluate ExpandSearch on **Medical RAG-QA datasets**, including MedQA-US, MedMCQA, PubMedQA, BioASQ-Y/N, and MMLU-Med. The retrieval corpus comprises Wikipedia, PubMed, and medical textbooks. We use E5 as the embedding model and report EM scores:
>
> |              | Model               | MedQA-US | MedMCQA | PubMedQA | BioASQ-Y/N | MMLU-Med | Avg. |
> | ------------ | ------------------- | -------- | ------- | -------- | ---------- | -------- | ---- |
> | s3           | 7B+Claude Haiku     | 45.7     | 45.4    | 13.6     | 6.5        | 56.2     | 33.5 |
> | ExpandSearch | 3B+Llama 4 Maverick | 8.8      | 8.2     | 68.8     | 84.8       | 10.0     | 36.1 |
>
> The contrasting performance patterns stem primarily from **architectural differences in the answer generation component**. S3 employs **Claude**, a commercial LLM with strong reasoning capabilities, as its answer generator, which explains its superior performance on complex multiple-choice tasks (MedQA-US, MedMCQA, MMLU-Med) that demand sophisticated medical reasoning. In contrast, ExpandSearch excels on evidence aggregation tasks (PubMedQA, BioASQ-Y/N) where comprehensive retrieval coverage through query expansion proves more beneficial.
>
> **Key Insight:** **ExpandSearch can be combined with S3** work by replacing our open-source squeezer with Claude or other commercial models, and also setting the answer generator to commercial models. This would leverage ExpandSearch's superior query expansion and evidence coverage while benefiting from commercial LLMs' advanced reasoning capabilities, potentially achieving the best of both approaches across all task types. We will explore this integration in future work.
>
> ---
>
> **References**
>
> [1] Gul, Mustafa Omer, Claire Cardie, and Tanya Goyal. "Pay-Per-Search Models are Abstention Models." arXiv preprint arXiv:2510.01152 (2025).

---

### Official Review · Reviewer_pzbi · 2025-10-31

**Soundness:** 3
**Presentation:** 3
**Contribution:** 1
**Rating:** 2
**Confidence:** 4

**Summary:**

This paper presents ExpandSearch, a reinforcement-learning-based methodology that trains the LLM agent to learn how to answer queries by asking multiple good questions to the search engine and perform retrieval-augmented generation. Compared to Search_R1, ExpandSearch improves the diversity of questions by generating multiple variants, which leads to higher recall during the retrieval stage. After this, all documents are summarized by another frozen LLM to ensure the conciseness of information. The approach is conceptually reasonable and provides effective performance gain on a variety of QA datasets.

**Strengths:**

1. The paper is generally well-written and easy to follow.
2. The idea of having a learnable strategy of writing multiple candidates and compressing the information is clean and reasonable.
3. Empirically, ExpandSearch outperforms many recent baselines on General QA and Multi-Hop QA using a variety of Qwen models.
4. The ablation study shows that each introduced component leads to ca ertain performance gain.

**Weaknesses:**

1. The method is a straightforward extension of Search-R1 by incorporating multiple queries and hence has limited novelty and insightfulness.
2. Using another LLM as a squeezer may introduce unwanted computational overhead, or even giving abilities that the small model does not possess.
3. Though the intuition of learning to write multiple diverse queries is beneficial, there is a lack of sufficient qualitative and quantitative analysis on the advantage brought by this strategy, other than the ablation study.

**Questions:**

1. There is a significant performance drop when the squeezer is removed, as shown in the ablation study. When using another larger LLM from a completely different model family (Llama instead of Qwen) as the squeezer, how to make sure that these models are not using their internal knowledge base to help answer the question, which may potentially cause information leakage?
2. Why is LLama chosen as the squeezer model instead of Qwen?
3. It would be beneficial to test the scaling of the number of queries and see when it roughly reaches the plateau, as an increasing number of queries is improving the performance almost linearly.

**Details Of Ethics Concerns:**

The paper’s writing is shockingly similar to ParallelSearch by Zhao et al, which is cited in the paper. The similarity in writing, graphics, and algorithms is sufficient to be considered plagiarism or self-plagiarism. It is also hard to clearly distinguish the contribution of this paper compared to ParallelSearch, although there are some expansions, which are unfortunately not highlighted. As a reviewer, I am very confused about the relationship between these two papers. Are they the same work or concurrent works? Regardless, I don’t think it complies with the paper publication standard in academia.

---

> ### Author Response · Authors · 2025-11-27
>
> We sincerely thank the reviewer for the thoughtful and constructive feedback. We greatly appreciate your recognition of the "_clean and reasonable idea_", "_easy to follow_" presentation, and "_strong results_." We address each of your valuable questions in detail below.
>
> ---
>
> > ***`Q1`: Novelty and Contributions***
>
> `A`: Thanks for your question. We propose a novel **architectural paradigm** for RL-based search agents. The core contribution is the **Expand-then-Squeeze** framework trained end-to-end with reinforcement learning.
>
> **Our work presents valuable insights for both academic research and industrial deployment:**
>
> 1. **Decoupling enables stable RL training:** By separating query generation (trained) from information extraction (frozen squeezer), we achieve stable optimization (`Figure 4`). In contrast, methods that jointly optimize all components suffer from instability (see training logs provided by Search-R1 github repo).
> 2. **Small query generators are sufficient:** Our 3B search agent with variable-size squeezers achieves comparable performance to 7B search agents (`Table 2`), significantly improving throughput, which is critical for datacenter deployment. Additionally, jointly trained search agents suffer from information overload, as demonstrated in `Table 3` and prior work.
> 3. **Training-free deployment flexibility:** `Figure 5` demonstrates plug-and-play squeezer selection, enabling a potential new paradigm where routers dynamically schedule squeezer size based on query difficulty without any retraining.
>
> ---
>
> > ***`Q2`: Computational Efficiency and Latency***
>
> `A`: We appreciate this concern regarding practical deployment considerations. **Counterintuitively, ExpandSearch achieves lower latency than baselines.**
>
> * 3B Baseline. The smaller model struggles to gather sufficient information, resulting in repeated query generation across multiple rounds, significantly increasing overall latency. This phenomenon is also documented in recent work such as [1].
> * 7B Baseline. While more capable, the larger model incurs higher per-token inference costs.
>
> **ExpandSearch's decoupled architecture** assigns the search task to an efficient 3B model while delegating extraction to a specialized squeezer, optimizing the compute-capability tradeoff.
>
> **Latency Comparison (ms per sample):**
>
> | Configuration                          | Latency (ms) |
> | -------------------------------------- | ------------ |
> | Search-R1 3B                           | 59.91        |
> | Search-R1 7B                           | 31.12        |
> | ExpandSearch (3B + Llama3-8B Squeezer) | **23.47**    |
>
>
> ---
>
> > ***`Q3`: Qualitative and Quantitative Analysis***
>
> `A`: Thanks for your question. We provide both qualitative and quantitative analyses:
> * **Qualitative:** Case studies are reported in `Section E (Appendix)`.
> * **Quantitative:** We demonstrate the effectiveness of the overall method compared to baselines (`Table 2`), the effect of the number of queries (`Figure 2`) and retrieved chunks (`Figure 3`), the effectiveness of individual components (`Table 3`), detailed expansion behavior analysis (`Table 4`), detailed squeezer behavior analysis (`Table 5`, `Figure 5`), and training dynamics (`Figure 4`).
>
> We welcome specific suggestions from the reviewer regarding additional experiments that would strengthen the analysis.
>
> ---
>
> > ***`Q4 & Q5`: Squeezer Model Family***
>
> `A`: Thanks for your question. We use LLaMA models because we relied on the NVIDIA NIM API as our model service provider, and Qwen models were not fully supported at the time of experimentation.
>
> Regarding information leakage: if a squeezer model possesses internal knowledge, we believe this can actually be a **beneficial signal** that helps extract more accurate information or correct errors in retrieved content (similar to how s3 [2] uses Claude to refine answers).
>
> To address this concern directly, we report results using **Qwen2.5-7B-Instruct** as the squeezer:
>
> | Squeezer   | NQ    | TriviaQA | PopQA | HotpotQA | 2wiki | Musique | Bamboogle | Avg.  |
> | ---------- | ----- | -------- | ----- | -------- | ----- | ------- | --------- | ----- |
> | LLaMA-3-8B | 0.449 | 0.634    | 0.452 | 0.398    | 0.406 | 0.175   | 0.500     | 0.431 |
> | Qwen2.5-7B | 0.430 | 0.629    | 0.454 | 0.403    | 0.425 | 0.172   | 0.518     | 0.433 |
>
> **Key Insight:** The comparable performance across different model families suggests that the gains primarily come from our framework design rather than model-specific knowledge leakage.

---

> ### Author Response · Authors · 2025-11-27
>
> > ***`Q6`: Scaling of the Number of Queries***
>
> `A`: We report results for different query counts to show the scaling behavior:
>
> |     | NQ    | TriviaQA | PopQA | HotpotQA | 2wiki | Musique | Bamboogle | Avg.  |
> | --- | ----- | -------- | ----- | -------- | ----- | ------- | --------- | ----- |
> | 1   | 0.385 | 0.616    | 0.337 | 0.332    | 0.318 | 0.090   | 0.347     | 0.340 |
> | 2   | 0.420 | 0.642    | 0.426 | 0.378    | 0.412 | 0.158   | 0.452     | 0.441 |
> | 3   | 0.446 | 0.677    | 0.456 | 0.422    | 0.450 | 0.194   | 0.540     | 0.457 |
> | 5   | 0.435 | 0.677    | 0.452 | 0.425    | 0.453 | 0.201   | 0.606     | 0.464 |
> | 7   | 0.425 | 0.680    | 0.447 | 0.410    | 0.465 | 0.170   | 0.568     | 0.452 |
>
> **Key Insight:** Performance improves substantially from 1 to 3 queries, with diminishing returns beyond that point. The plateau around 3–5 queries suggests a natural saturation point in query diversity benefits.
>
> ---
>
> > ***`Q7`: Relationship to ParallelSearch***
>
> `A`: While ParallelSearch also generates multiple queries, the **task goals, focus, solutions, and insights are fundamentally different**:
>
> | Aspect            | ParallelSearch                             | ExpandSearch (Ours)                                             |
> | ----------------- | ------------------------------------------ | --------------------------------------------------------------- |
> | **Goal**          | Reduce LLM calls via parallelization       | Improve information coverage and handle overload                |
> | **Focus**         | Identifying parallelizable sub-questions   | Generating semantically diverse query variants                  |
> | **Architecture**  | Unified search agent                       | Decoupled expand-then-squeeze framework                         |
> | **Reward Design** | Dedicated reward for decomposition quality | Vanilla reward to demonstrate simplicity                        |
> | **Key Insight**   | Query parallelization reduces latency      | Decoupling enables stable training and plug-and-play deployment |
>
> ParallelSearch decomposes questions into parallelizable sub-questions in a single-round interaction but **cannot address information coverage and overload issues**. Our ExpandSearch takes a fundamentally different perspective: encouraging LLMs to generate diverse queries while proposing a decoupled architecture that enables plug-and-play information compression. We intentionally use the vanilla reward function to demonstrate the simplicity of our approach and its ease of integration with existing methods.
>
> ---
>
> **References**
>
> [1] Gul, Mustafa Omer, Claire Cardie, and Tanya Goyal. "Pay-Per-Search Models are Abstention Models." arXiv preprint arXiv:2510.01152 (2025).
>
> [2] Jiang, Pengcheng, et al. "s3: You Don't Need That Much Data to Train a Search Agent via RL." EMNLP 2025.

---

### Official Review · Reviewer_5JKj · 2025-11-01

**Soundness:** 2
**Presentation:** 3
**Contribution:** 3
**Rating:** 6
**Confidence:** 4

**Summary:**

The paper introduces ExpandSearch, a dual-model RL framework that addresses limitations of single-query retrieval in reasoning-augmented LLMs like Search-R1. The expand stage trains an LLM to generate multiple semantically diverse queries (syntactic and semantic expansions) for broader evidence coverage. The squeeze stage uses a frozen summarization model (“squeezer”) to distill the retrieved documents into compact, relevant contexts and removes noise.

The approach is trained via RL based on Exact Match (EM) reward. Experiments on seven QA benchmarks (including HotpotQA, Musique, 2Wiki, Bamboogle, NQ, TriviaQA, PopQA) show EM gains over Search-R1 and related RL-based retrieval agents (ZeroSearch, ParallelSearch, Router-R1). The authors claim the method enables even a 3B model to outperform 7B baselines by improving query diversity and information precision.

**Strengths:**

1. Strong motivation and clear insight: The key observation that LLM-based search agents struggle with both semantic incompleteness (too narrow queries) and information overload (too much irrelevant retrieved text) is well articulated and supported.

2. The “expand-then-squeeze” decomposition is intuitively appealing and aligns with human search behavior.

3. Decoupling the retrieval (expansion) and summarization (squeeze) components is practical, scalable, and computationally efficient. The modularity allows plug-and-play squeezers without retraining.

4. The paper is easy to follow, with good examples and clear algorithmic description.

5. Results are consistent across datasets and model sizes. The ablation studies (with/without squeezer, with/without syntax or semantic expansion) reveal useful insights about the learned expansion behavior.

6. RL training seems stable

**Weaknesses:**

1.  The model is trained and evaluated using the same EM metric. This creates a bias, the agent is directly optimizing for the benchmark metric, so reported gains may simply reflect reward overfitting rather than genuine reasoning or retrieval improvements.

2. Other evaluation metrics like F1, LLM-as-judge, or match scores (e.g., Match, F1) are absent. Without those, the evaluation is incomplete.

3. The paper repeatedly claims that ExpandSearch “improves recall” and “balances recall and precision,” but does not provide any retrieval metrics. Without retrieval-level analysis, it’s unclear whether gains come from better retrieval, better summarization, or just reformulations.

4. While the paper includes seven datasets, only 2-4 hop QA benchmarks are used (HotpotQA, 2Wiki, Musique, Bamboogle). These are relatively shallow multi-hop tasks by current standards.

5. The authors omit comparisons with recent strong baselines that report state of the art performance on multi-hop rag benchmarks, including: CoRAG (Wang et al), R1-Searcher (Song et al), FrugalRAG (Java el al), O2-Searcher (Mei et al), etc. The squeeze stage is conceptually similar to RECOMP (Xu et al.), which also compresses retrieved text using a summarizer before answer generation. Similarly, it would be good to make clear distinction with existing methods that use search expansion. ExpandSearch differs in using RL for multi-query expansion, but both share the same underlying goal of mitigating context overload by selective compression.

6. The approach seems conceptually similar to prior works mentioned in the paper like Router-R1, ParallelSearch, which also use multiple queries. A clear distinction would position this work better.

7. The authors claim cross-domain generalization, but nearly all benchmarks (NQ, TriviaQA, Hotpot, etc.) are Wikipedia-derived, sharing similar style and domain distribution. This is a weak claim without including additional, real world datasets (e.g. biology).

8. The paper claims even a small 3B model shows strong performance, but that’s achieved only by offloading summarization to a larger frozen model. The overall system compute footprint is much greater than single-model baselines. A clear comparison of latency, overall model size (including squeezer) is required for a fair comparision

**Questions:**

Kindly see weaknesses.

---

> ### Author Response · Authors · 2025-11-27
>
> We sincerely thank the reviewer for the thoughtful and constructive feedback. We greatly appreciate your recognition of our work's "_strong motivation and clear insight_," "_easy to follow_" presentation, and "_strong results_." Below, we address each concern in detail.
>
> ---
>
> > ***`Q1 & Q2`: Evaluation Metrics***
>
> `A`: We appreciate this important methodological concern. Following the experimental protocol established by Search-R1, we adopt identical settings, datasets, and metrics to ensure fair comparison. Notably, our evaluation spans **seven benchmarks including unseen datasets**, which specifically tests generalization beyond the training distribution and mitigates overfitting concerns.
>
> To provide a more comprehensive evaluation, we report additional metrics below:
>
> **F1 Score:**
>
> |              | NQ    | TriviaQA | PopQA | HotpotQA | 2wiki | Musique | Bamboogle | Avg.  |
> | ------------ | ----- | -------- | ----- | -------- | ----- | ------- | --------- | ----- |
> | Search-R1    | 0.453 | 0.620    | 0.434 | 0.418    | 0.373 | 0.171   | 0.386     | 0.408 |
> | ExpandSearch | 0.547 | 0.755    | 0.500 | 0.541    | 0.524 | 0.284   | 0.695     | 0.549 |
>
> **LLM-as-a-Judge Accuracy** (using prompts from s3 [1]):
>
> |              | NQ    | TriviaQA | PopQA | HotpotQA | 2wiki | Musique | Bamboogle | Avg.  |
> | ------------ | ----- | -------- | ----- | -------- | ----- | ------- | --------- | ----- |
> | Search-R1    | 0.507 | 0.751    | 0.466 | 0.536    | 0.421 | 0.328   | 0.336     | 0.478 |
> | ExpandSearch | 0.656 | 0.869    | 0.533 | 0.708    | 0.703 | 0.467   | 0.632     | 0.653 |
>
> **Key Insight:** These results demonstrate consistent improvements across multiple evaluation paradigms, supporting the robustness of our findings beyond the training metric.
>
> ---
>
> > ***`Q3`: Retrieval-Level Analysis for Recall and Precision Claims***
>
> `A`: We appreciate this request for deeper analysis. Both query expansion and the squeezer contribute to performance improvements, as demonstrated in `Figure 2` and `Table 3`:
> * `Figure 2`: ExpandSearch outperforms baselines even when query expansion is disabled, demonstrating the squeezer's effectiveness in improving information **precision**.
> * `Table 3`: ExpandSearch outperforms baselines without a squeezer, demonstrating query expansion's effectiveness in improving information **recall**.
>
> To directly quantify recall improvements, we employ the recall metric defined in [2] (a simplified information coverage measure):
>
> |              | NQ    | TriviaQA | PopQA | HotpotQA | 2wiki | Musique | Bamboogle | Avg.  |
> | ------------ | ----- | -------- | ----- | -------- | ----- | ------- | --------- | ----- |
> | Search-R1    | 0.442 | 0.659    | 0.409 | 0.565    | 0.523 | 0.252   | 0.216     | 0.508 |
> | ExpandSearch | 0.636 | 0.802    | 0.551 | 0.617    | 0.706 | 0.351   | 0.528     | 0.650 |
>
> **Key Insight:** This retrieval-level analysis confirms that our gains stem from genuinely improved information coverage rather than merely better reformulations.
>
> ---
>
> > ***`Q4`: Multi-hop Task Complexity***
>
> `A`: Thanks for your suggestion. We strictly follow established search agent baselines (e.g., **Search-R1**, **s3**) in our experimental setup to ensure fair and reproducible comparisons. While we acknowledge that deeper multi-hop benchmarks exist, our primary goal is to demonstrate improvements over comparable methods under identical evaluation conditions. We will consider incorporating more complex reasoning benchmarks in future work.
>
> ---
>
> > ***`Q5`: Comparison with Additional Baselines***
>
> `A`: Thanks for your suggestion. We acknowledge the rapidly evolving landscape of RL-based search agents / RAG, with diverse settings, training methodologies, model sizes, and data curation pipelines. Given resource constraints, exhaustive comparison under identical conditions is infeasible. We therefore build upon Search-R1, **a strong peer-reviewed baseline**, to ensure fair evaluation and select comparison methods with similar experimental settings.
>
> Regarding RECOMP: while it is an influential work in RAG compression, ExpandSearch differs in two key aspects:
> - **RL-based optimization:** Our method leverages reinforcement learning to enhance the LLM's query generation and reasoning capabilities.
> - **Plug-and-play architecture:** More importantly, ExpandSearch enables modular deployment of the squeezer (compressor), offering flexibility beneficial for both academic research and industrial applications.

---

> ### Author Response · Authors · 2025-11-27
>
> > ***`Q6`: Distinction from Router-R1 and ParallelSearch***
>
> `A`: Thanks for your question. While these methods share the surface-level characteristic of generating multiple queries, they address fundamentally different challenges:
>
> |Method|Primary Focus|Key Mechanism|
> |---|---|---|
> |**Router-R1**|Dynamic routing & cost control|Generates multiple queries and selects the most appropriate LLM for answering|
> |**ParallelSearch**|Reducing LLM calls|Identifies parallelizable sub-questions for single-round decomposition|
> |**ExpandSearch**|Information coverage & overload mitigation|Encourages diverse query generation to maximize information coverage while managing context length|
>
> Crucially, neither Router-R1 nor ParallelSearch addresses the dual challenges of **information coverage** and **context overload** that ExpandSearch specifically targets.
>
> ---
>
> > ***`Q7`: Cross-Domain Generalization***
>
> `A`: Thanks for your question. Following s3, we evaluate ExpandSearch on **Medical RAG-QA datasets**, including MedQA-US, MedMCQA, PubMedQA, BioASQ-Y/N, and MMLU-Med. The retrieval corpus comprises Wikipedia, PubMed, and medical textbooks. We use E5 as the embedding model and report EM scores:
>
> |              | Model               | MedQA-US | MedMCQA | PubMedQA | BioASQ-Y/N | MMLU-Med | Avg. |
> | ------------ | ------------------- | -------- | ------- | -------- | ---------- | -------- | ---- |
> | s3           | 7B+Claude Haiku     | 45.7     | 45.4    | 13.6     | 6.5        | 56.2     | 33.5 |
> | ExpandSearch | 3B+Llama 4 Maverick | 8.8      | 8.2     | 68.8     | 84.8       | 10.0     | 36.1 |
>
> The contrasting performance patterns stem primarily from **architectural differences in the answer generation component**. S3 employs **Claude**, a commercial LLM with strong reasoning capabilities, as its answer generator, which explains its superior performance on complex multiple-choice tasks (MedQA-US, MedMCQA, MMLU-Med) that demand sophisticated medical reasoning. In contrast, ExpandSearch excels on evidence aggregation tasks (PubMedQA, BioASQ-Y/N) where comprehensive retrieval coverage through query expansion proves more beneficial.
>
> **Key Insight:** **ExpandSearch can be combined with S3** work by replacing our open-source squeezer with Claude or other commercial models, and also setting the answer generator to commercial models. This would leverage ExpandSearch's superior query expansion and evidence coverage while benefiting from commercial LLMs' advanced reasoning capabilities, potentially achieving the best of both approaches across all task types. We will explore this integration in future work.
>
>
> ---
>
> > ***`Q8`: Computational Efficiency and Latency***
>
> `A`: We appreciate this concern regarding practical deployment considerations. **Counterintuitively, ExpandSearch achieves lower latency than baselines.**
>
> * 3B Baseline. The smaller model struggles to gather sufficient information, resulting in repeated query generation across multiple rounds, significantly increasing overall latency. This phenomenon is also mentioned in recent work such as [3].
> * 7B Baseline. While more capable, the larger model incurs higher per-token inference costs.
>
> **ExpandSearch's decoupled architecture** assigns the search task to an efficient 3B model while delegating extraction to a specialized squeezer, optimizing the compute-capability tradeoff.
>
> **Latency Comparison (ms per sample):**
>
> | Configuration                          | Latency (ms) |
> | -------------------------------------- | ------------ |
> | Search-R1 3B                           | 59.91        |
> | Search-R1 7B                           | 31.12        |
> | ExpandSearch (3B + Llama3-8B Squeezer) | **23.47**    |
>
> ---
>
> **References**
>
> [1] Jiang, Pengcheng, et al. "s3: You Don't Need That Much Data to Train a Search Agent via RL." EMNLP 2025.
>
> [2] Wang, Yiding, et al. "Beyond Outcome Reward: Decoupling Search and Answering Improves LLM Agents." arXiv preprint arXiv:2510.04695 (2025).
>
> [3] Gul, Mustafa Omer, Claire Cardie, and Tanya Goyal. "Pay-Per-Search Models are Abstention Models." arXiv preprint arXiv:2510.01152 (2025).

---

### Official Review · Reviewer_Qimr · 2025-11-03

**Soundness:** 2
**Presentation:** 3
**Contribution:** 1
**Rating:** 0
**Confidence:** 5

**Summary:**

The paper proposes an extension to the Search-R1 framework by including a query expansion component for retrieval. The model does multiple query expansions simultaneously and is trained using RL. Comparison against Search-R1 shows about 4% performance gain.

**Strengths:**

+ The paper focuses on a timely topic and Search-R1 is a great platform to explore this idea.
+ The paper evaluates on seven datasets.
+ The approach is simple and the paper is easy to follow.

**Weaknesses:**

- The first main weakness is comparison with weak baselines. Even though the authors extended Search-R1 with query expansion, they should have compared the model with Search-R1 that interacts with a search engine capable of query expansion. Query expansion in retrieval has been studied for decades. There are many methods, such as Relevance Models, Embedding-based Query Expansion techniques, and even LLM-based query expansions that do not need any training. Search-R1 with these query expansion methods should be considered as baselines. Otherwise, it's unclear where the gains come from. Are they coming from the fact that query expansion is useful for retrieval? (which is a known phenomenon) or does the proposed approach for query expansion work compared to existing alternatives?

- The presented results are confusing. How come results for Search-R1 and ExpandSearch in Table 3 don't match with any of the Search-R1 and ExpandSearch results in Table 2? By the way, the ablation study should be done on either all LLM sizes or on the Qwen base 7B where the difference between Search-R1 and ExpandSearch is the smallest.

- No statistical significance tests are done to demonstrate the obtained improvements are meaningful.

- The main contribution of this work is on query expansion, which is basically an approach for improved retrieval (not language modeling). Therefore, the focus on the experiments should be more on various retrieval models and query expansion baselines instead of blindly following the trend on experimenting with different LLM sizes. For example, does this query expansion approach work if they retrieval model is based on term matching, like BM25 or KL divergence? What about other dense retrieval models, like DPR and ColBERT? What about retrieval models with sparse representations, like SPLADE? What about more advanced retrieval models, like Hypencoder?

- Analysis on the impact of the number of expanded tokens is missing.

- Last but not least, the paper lacks sufficient novelty for ICLR. Seems just like a prompt engineering paper that encourages the model to perform query expansion, rather than studying how to design a reward functions for this task, how to optimize this effectively, and so on.

**Questions:**

None. The concerns raised in the Weakness section are so significant that cannot be resolved in a rebuttal.

---

> ### Author Response · Authors · 2025-11-27
>
> We sincerely thank the reviewer for the thoughtful and constructive feedback. We greatly appreciate your recognition that we are addressing a "_timely topic_" and that the proposed method is "_easy to follow_". We address each of your valuable questions in detail below.
>
> ---
>
> > ***`Q1`: Query Expansion Baselines***
>
> `A`: Thanks for your question. We have compared ExpandSearch with Search-R1 using a direct expansion prompt baseline in `Table 3`. The results show that **simply adding the expansion mechanism without training yields no improvement and even slightly degrades performance.**
>
> To further support this claim, we conducted additional experiments with established query expansion methods:
>
> **Experimental Setup**
>
> - **LLM:** `Search-R1-Qwen2.5-3B-Instruct`
> - **Query Expansion Methods:** `RM3`, `HyDE`, `ColBERT-PRF`
> - **Hyperparameters:** Same as reported in the main paper
>
> **Results**
>
> |Method|NQ|TriviaQA|PopQA|HotpotQA|2wiki|Musique|Bamboogle|Avg.|
> |---|---|---|---|---|---|---|---|---|
> |Search-R1|0.341|0.545|0.378|0.324|0.319|0.103|0.264|0.325|
> |Search-R1 + RM3|0.374|0.640|0.361|0.368|0.381|0.142|0.396|0.380|
> |Search-R1 + HyDE|0.410|0.627|0.374|0.345|0.331|0.142|0.395|0.375|
> |Search-R1 + ColBERT-PRF|0.347|0.615|0.377|0.332|0.304|0.123|0.347|0.349|
> |**ExpandSearch (Ours)**|**0.446**|**0.677**|**0.456**|**0.422**|**0.450**|**0.194**|**0.540**|**0.457**|
>
> **Key Insight:** End-to-end RL training enables the model to generate more effective and diverse queries, substantially outperforming traditional query expansion baselines.
>
> ---
>
> > ***`Q2`: Results in Table 3***
>
> `A`: Thanks for your question. For Search-R1, as explained in `Lines 365–366`, we set the number of retrievals to `15` since ExpandSearch uses `3 × 5 = 15` retrievals. For ExpandSearch, we reran the evaluation script, which may yield slightly different results within a small margin. We will use the results from Table 2 directly in the final version to ensure consistency.
>
> We also report an ablation study based on the **7B base model** as requested:
>
> **Experimental Setup**
>
> - **LLM:** `Qwen2.5-7B-Base`
> - **Number of Retrievals:** `15`
> - **Hyperparameters:** Same as reported in the main paper
>
> **Results**
>
> |Method|NQ|TriviaQA|PopQA|HotpotQA|2wiki|Musique|Bamboogle|Avg.|
> |---|---|---|---|---|---|---|---|---|
> |ExpandSearch|0.496|0.703|0.506|0.445|0.488|0.196|0.540|0.480|
> |w/o squeezer|0.468|0.671|0.477|0.413|0.457|0.188|0.496|0.453|
> |Search-R1|0.480|0.638|0.457|0.433|0.382|0.196|0.432|0.431|
> |w/ Expansion + Squeezer|0.485|0.651|0.462|0.428|0.394|0.203|0.450|0.439|
>
> **Key Insight:** The expand-then-squeeze paradigm maintains its effectiveness when scaling up the base model.
>
> * **
>
> > ***`Q3`: Significance Tests***
>
> `A`: Thanks for your suggestion. We conducted comprehensive statistical significance tests on **51,713 paired samples**. All tests confirm that ExpandSearch's improvements over Search-R1 are statistically significant (_p_ < 0.001). Notably, ExpandSearch correctly answers **3.6× more** questions that Search-R1 fails on (9,422 vs. 2,653), demonstrating consistent and substantial improvement rather than random variation.
>
> |Test|Result|
> |---|---|
> |Paired t-test|_p_ < 0.001|
> |Wilcoxon signed-rank test|_p_ < 0.001|
> |Bootstrap 95% CI (accuracy gain)|[+12.7%, +13.5%]|
> |McNemar's test|_p_ < 0.001|
>
> ---
>
> > ***`Q4`: More Retrievers***
>
> `A`: Thanks for your question. We report results using both **sparse** and **dense** retrievers, aligned with the Search-R1 experimental setup:
>
> **Experimental Setup**
>
> - **LLM:** `Qwen2.5-3B-Instruct`
> - **Retrievers:** `BM25` (Sparse), `E5` (Dense), `ColBERT` (Dense)
> - **Number of Retrievals:** `15`
> - **Hyperparameters:** Same as reported in the main paper
>
> **Results**
>
> | Retriever | NQ    | TriviaQA | PopQA | HotpotQA | 2wiki | Musique | Bamboogle | Avg.  |
> | --------- | ----- | -------- | ----- | -------- | ----- | ------- | --------- | ----- |
> | BM25      | 0.428 | 0.670    | 0.406 | 0.413    | 0.470 | 0.191   | 0.480     | 0.437 |
> | E5        | 0.446 | 0.677    | 0.456 | 0.422    | 0.450 | 0.194   | 0.540     | 0.457 |
> | ColBERT   | 0.428 | 0.655    | 0.508 | 0.407    | 0.453 | 0.218   | 0.589     | 0.466 |
>
> **Key Insight:** These results confirm the findings in Search-R1: training with stronger retrievers yields better final performance. Our method generalizes across retriever architectures.
>
> **Please note that the different retrievers can be viewed as an ability for tool use in RL agents, and this is not the main point of our work or of recent RL search agent works.**
>
> ---
>
> > ***`Q5`: Analysis on the Number of Expansion***
>
> `A`: Thanks for your question. We have reported this analysis in `Figure 2` and `Figure 3` of the main paper.

---

> ### Author Response · Authors · 2025-11-27
>
> > ***`Q6`: Novelty and Contributions***
>
> `A`: Thanks for your question. However, we respectfully believe the reviewer may have **misunderstood the key challenges in RL-based search agents and our contributions**.
>
> **First, our method is NOT a prompt engineering paper.** We propose a novel **architectural paradigm** for RL-based search agents. The core contribution is the **Expand-then-Squeeze** framework trained end-to-end with reinforcement learning. Please refer to `Lines 097–107` for detailed contributions.
>
> **Second, our work presents valuable insights for both academic research and industrial deployment:**
>
> 1. **Decoupling enables stable RL training:** By separating query generation (trained) from information extraction (frozen squeezer), we achieve stable optimization (`Figure 4`). In contrast, methods that jointly optimize all components suffer from instability (see training logs provided by Search-R1 github repo).
> 2. **Small query generators are sufficient:** Our 3B search agent with variable-size squeezers achieves comparable performance to 7B search agents (`Table 2`), significantly improving throughput, which is critical for datacenter deployment. Additionally, jointly trained search agents suffer from information overload, as demonstrated in `Table 3` and prior work.
> 3. **Training-free deployment flexibility:** `Figure 5` demonstrates plug-and-play squeezer selection, enabling a potential new paradigm where routers dynamically schedule squeezer size based on query difficulty without any retraining.

---

### Author Response · Authors · 2025-11-29

We sincerely appreciate all reviewers for their insightful and constructive feedback. We extend our special gratitude to ACs/SACs/PCs for their tremendous efforts, particularly during this period of significantly increased workload.

---

***Summary***

We present ExpandSearch, a reinforcement learning framework that trains LLM-based search agents with native query expansion capabilities. Our core innovation is the Expand-then-Squeeze architecture: (1) an RL-trained agent generates multiple semantically diverse queries for broader evidence coverage, and (2) a frozen squeezer model compresses retrieved documents to mitigate information overload. Our method achieves 4.4% average improvement over state-of-the-art baselines. This simple yet effective decoupled architecture provides valuable insights for both academic research and industrial deployment: stable RL training through decoupling, demonstrating that small query generators are sufficient, and enabling flexible, training-free deployment. During rebuttal, we conducted extensive additional experiments addressing all reviewer concerns.

---

***Key Concerns and Our Responses***

1. **Compared to traditional information retrieval methods (Reviewer `Qimr`)**

**Concern**: What is the performance compared with traditional query expansion methods and using different retrievers?

**Response**: We conducted new experiments comparing ExpandSearch against established query expansion methods integrated with Search-R1 and reported results using both sparse and dense retrievers.

**Key Insight**:  RL-trained query expansion in ExpandSearch yields substantial improvements compared to Traditional query expansion methods, and Stronger retrievers achieve a higher performance in ExpandSearch.

2. **Evaluation Metrics Beyond EM (Reviewer `5JKj`)**

**Concern**: What is the F1 score or LLM-as-a-Judge accuracy?

**Response**: We provide comprehensive F1 scores and LLM-as-a-Judge accuracy.

**Key Insight**: ExpandSearch still outperforms baselines using different evaluation metrics.

3. **Cross-Domain Generalization (Reviewers `5JKj`, `Mg5C`)**

**Concern**: What is the performance when applying ExpandSearch for other domains?

**Response**: We evaluated on Medical RAG-QA datasets (MedQA-US, MedMCQA, PubMedQA, BioASQ-Y/N, MMLU-Med).

**Key Insight**: ExpandSearch excels on evidence aggregation tasks where comprehensive retrieval coverage proves beneficial. The method can be combined with commercial LLMs for complex reasoning tasks.

4. **Distinction from Related Work (Reviewers `5JKj`, `pzbi`)**

**Concern**: How does ExpandSearch differ from several previous works?

**Key Distinction**: ExpandSearch uniquely addresses both information coverage (through RL-trained query expansion) and context overload (through modular squeezer) simultaneously, bringing valuable insights for both academic research and industrial deployment.

---

***Additional Rebuttal Contributions***

* **Query Scaling Analysis**: Performance scales with the number of queries.
* **Squeezer Model Family**: Comparable results across LLaMA and Qwen squeezers, confirming gains from framework design rather than knowledge leakage.
* **Statistical Significance**: Performance differences are statistically significant.
* **Recall Metric**: ExpandSearch outperforms baselines regarding the recall ability.
* **Computational Efficiency and Latency**: ExpandSearch achieves lower latency than baselines.

---

***Conclusion***

We believe our comprehensive rebuttal addresses all reviewer concerns with concrete experimental evidence.

---

### Meta-Review · Area_Chair_YpcK · 2025-12-21

**Summary:**

This paper proposed ExpandSearch, a search-augmented agent that first performs query expansion during the search step to increase the coverage of retrieved information, and then uses a “squeezer” that summarizes the search results into high-density information for the agent. Results on a range of general QA and multi-hop QA datasets show substantial improvements over baselines such as Search-R1 and Router-R1.

Concerns that are resolved
- Using EM during training introduces bias; missing additional metrics like F1, LLM-as-a-judge (5JKj): Addressed.
- Comparison to more recent work (5JKj): The author’s justification for using Search-R1 as a strong peer-reviewed baseline makes sense.
- Significance tests, more retrieval models, baseline results reported in the paper not matching with the original papers (Qimr): The rebuttal sufficiently addressed the concern.

Concerns that may or may not have been resolved
- Computational efficiency and latency (pzbi, Mg5C, 5JKj): The authors provide latency breakdown, suggesting that small models are actually slower due to requiring more iterations. I found this conclusion slightly confusing, and the trade-offs between iteration count, accuracy, and latency still remains unclear. Additionally, involving a larger model as part of the system would introduce costs beyond latency, such as memory requirement, which are not fully discussed.
- No cross-domain generalization (Mg5C, 5JKj): Additional experiments are reported, although I’m not sure if the reviewer would be convinced, because it lacks controlled experiments (e.g., only compared with s3 that uses a totally different LM).
- Only 2–4 hop QA benchmarks are used which are shallow (5JKj): I share the concern with the reviewer. At the same time, the authors’ rebuttal makes sense that most prior search agent papers have already been using these datasets as standard benchmarks, so it is a reasonable choice for authors to follow that. More broadly, I do agree with the reviewer the the field should move away from these shallow, 2–4 hope QA benchmarks.

The following concerns were raised by reviewers and also align with the AC’s independent assessment (formed prior to reading reviews), and, in my opinion, not fully resolved by the rebuttal.
- The proposed work is a fairly straightforward extension of Search-R1; it is conceptually similar to other prior work (there are differences, but there’re minor), and has limited novelty and insightfulness (pzbi, 5JKj, Qimr). The AC’s independent judgement agrees with these assessments and the AC does not think the authors’ rebuttal sufficiently addressed this.
- Writing quality (5JKj): The paper’s writing can improve, especially with respect to repeated claims about the methods, e.g., “improved recall” and “balancing recall and precision” without providing clear qualitative or mechanistic evidence to support these claims.
- Missing literature (Qimr): The AC’s independent judgement agrees that the paper is dismissing a large body of existing literature in query expansion. Beyond older, classic work, it also appears to miss recent search-agent papers with closely related ideas, such as search-o1, which has a component that is almost equivalent to the “squeezer” proposed in this paper.

**Reviewer Concerns:**

(Noted above)

**Reviewer Scores:**

(Noted above)

---

### Decision · Program_Chairs · 2026-01-26

Reject